# Association between the Cognitive-Related Behavioral Assessment Severity Stage and Activities of Daily Living Required for Discharge to Home in Patients with Stroke: A Cross-Sectional Study

**DOI:** 10.3390/ijerph20043005

**Published:** 2023-02-09

**Authors:** Yoshiaki Maki, Akiko Morita, Hyuma Makizako

**Affiliations:** 1Department of Rehabilitation, Ukai Rehabilitation Hospital, Nagoya 453-0811, Japan; 2Department of Physical Therapy, School of Health Sciences, Faculty of Medicine, Kagoshima University, Kagoshima 890-0065, Japan

**Keywords:** stroke, cognitive function, cognitive-related behavioral assessment, activities of daily living, cross-sectional study

## Abstract

This study aimed to characterize cognitive function examined using Cognitive-related Behavioral Assessment (CBA) in activities of daily living (ADLs). According to CBA severity at discharge, 791 patients were assigned to five groups (most severe, severe, moderate, mild, and normal). The total scores for Functional Independence Measure (FIM) motor items were compared for each group. Multiple logistic regression analysis was performed to clarify the association between CBA severity and independence in ADL items. Independence in each ADL according to CBA severity was 0–4.8%, 26.8–45.0%, 84.3–91.0%, and 97.2–100% for all ADLs in the most severe to severe, moderate, mild, and normal groups, respectively. Significant differences were found in the FIM motor score according to CBA severity between the groups (*p* < 0.01). A mild or normal CBA was associated with a higher odds ratio (OR) for dressing the upper body (OR = 21.90; 95% confidence interval (CI), 13.50–35.70), bladder management (OR = 11.60; 95% CI, 7.21–18.60), transfers to the bed/chair/wheelchair (OR = 18.30; 95% CI, 11.40–29.40), transfers to the toilet (OR = 18.30; 95% CI, 11.40–29.30), and walking (OR = 6.60; 95% CI, 10.60–26.10). A CBA severity greater than mild (23 points) was associated with independence in ADLs that are important for discharge to home.

## 1. Introduction

Poststroke cognitive impairment is present in 22–85% of patients with stroke in the subacute phase [1,2,3,4]. Previous studies have shown that poststroke cognitive impairment affects rehabilitation outcomes, independence in activities of daily living (ADLs) and walking, and functional improvement [4,5,6]. Additionally, cognitive impairment increases the risk of falls [7]. National and international rehabilitation guidelines recommend that these symptoms be considered [8,9]. The Canadian Stroke Best Practice Recommendations suggest considering the risk of poststroke cognitive impairment in all patients with stroke [8].

Generally, poststroke cognitive impairment is examined using paper-based neuropsychological tests, such as the Mini-Mental State Examination (MMSE) [10], Raven’s Colored Progressive Matrices (RCPM) [11], and the Behavioral Inattention Test [12]. These tests objectively capture the severity of poststroke cognitive impairment and are useful in detecting and diagnosing poststroke cognitive impairment [13]. However, neuropsychological testing alone is limited in its ability to accurately detect disorders due to impaired consciousness and aphasia [14,15], and testing is difficult in cases of severe impairment of arousal, emotion, and communication, thereby limiting the target population. Additionally, poststroke cognitive impairment detected in ADL situations is difficult to assess by neuropsychological testing [16]. It is not sufficient to determine the risk of cognitive impairment [8] in all the aforementioned patients with stroke, and it may be difficult to measure the cognitive functions required for ADLs directly. Therefore, it could be useful to examine the effects of poststroke cognitive impairment on ADL performance by observing the behavior of patients with stroke in their daily lives [17,18]. Behavioral observation tests, such as the Catherine Bergego Scale [17] and Moss Attention Rating Scale [18], have been reported to be beneficial. However, these assessments only evaluate limited types of poststroke cognitive impairment, hemispatial neglect, and attention impairment and are not sufficient for evaluating recovering stroke patients with multiple overlapping poststroke cognitive impairments. Recently, there have been much-needed reports of comprehensive observational assessments of poststroke cognitive impairment [19].

The Cognitive-related Behavioral Assessment (CBA) [20] was developed to evaluate cognitive function-related problems that occur in patients with stroke and has been found useful in clinical practice. The CBA is based on the concepts of the neuropsychological pyramid [21] and the behavioral and cognitive impairment model [22], which clearly illustrate the hierarchy of general symptoms of poststroke cognitive impairment and target the evaluation of general symptoms that are difficult to assess by neuropsychological tests. Previous studies have reported that the CBA has good to excellent inter-rater reliability and internal consistency [20,23]. Reports on the comorbid validity of the CBA show high correlations with the MMSE and the RCPM, which are used for cognitive impairment [20]. Regarding ecological validity, a strong correlation between the CBA and ADLs has also been observed [20]. We also proposed the use of the CBA to determine the stage of severity of illness and to assist patients with stroke according to the severity of illness [24]. However, it is unclear how the severity of cognitive function assessed by the CBA is characterized in terms of ADLs [25], which are necessary for discharge to home. More clarity is needed to relate the characteristics of ADL abilities to CBA severity scores. In particular, the severity of ADLs that are important for discharge to home can provide clues for setting rehabilitation goals and the rehabilitation content required by therapists.

Therefore, this study aimed to clarify ADL characteristics according to severity, as judged by the CBA total score, in patients with stroke admitted to rehabilitation wards and to examine the impact of CBA severity on more important ADL items for discharge to home.

## 2. Materials and Methods

### 2.1. Ethical Considerations

The study was approved by the Ethics Committee of Ukai Rehabilitation Hospital (approval number: 2022-0020). In addition, information about this study was published on the Ukai Rehabilitation Hospital website, and an opt-out procedure was used to allow patients and their families to refuse participation in this study if needed.

### 2.2. Study Design and Patients

This was a single-center cross-sectional study. Patients admitted to the recovery phase rehabilitation wards of Ukai Rehabilitation Hospital between October 2017 and March 2022 were included. Patients with first-episode stroke admitted for rehabilitation, those hospitalized for at least 1 month, and those assessed on admission and discharge were included in the study. Patients with pre-existing cerebrovascular disease or bilateral motor paralysis were excluded. Patients who were transferred or readmitted to the hospital and those who did not complete the assessment items were also excluded (Figure 1). In total, 791 patients were included in the analysis after excluding those who met the exclusion criteria. In addition to assessment at admission and discharge, patients were assessed every month after admission during their hospital stay. All patients in this study underwent rehabilitation programs administered by physical, occupational, and speech therapists. Rehabilitation programs were based on a comprehensive approach and included physical, occupational, and speech therapies. Patients were provided with 6–9 units (1 unit: 20 min) per day, as necessary.

### 2.3. Measurement Items

The evaluation results of the patients at the time of discharge were extracted from the usual medical data. Evaluations at discharge were conducted in the days prior to discharge.

#### 2.3.1. Patient Attributes

Data on age, sex, type of stroke (cerebral infarction, cerebral hemorrhage, or subarachnoid hemorrhage), hemisphere of stroke, days since stroke onset, and Brunnstrom Recovery Stage (BRS) were extracted.

#### 2.3.2. Cognitive-Related Behavioral Assessment (Table A1)

Even patients with aphasia and unilateral spatial neglect are not affected by their symptoms, and we aimed to capture a comprehensive overall cognitive function that includes them. The CBA is performed by observing the patient’s behavior in daily life and in hospital [20]. The viewpoints of the observation items are defined based on six categories of consciousness, emotion, attention, memory, judgment, and consciousness of disease, and each item is rated on a 5-point scale: 5, normal; 4, mild; 3, moderate; 2, severe; and 1, most severe (Appendix A). For example, in the evaluation item, “consciousness,” sleep and wakefulness rhythms are stable, and mental energy can be sustained without showing moderate fatigue or decreased response during cognitive activities. From this viewpoint, behavioral observations were made, and the patients were fully awake and not tired (5 points), generally awake but occasionally vague (4 points), vague from beginning to end (3 points), and occasionally showed a tendency to somnolence (2 points), and always somnolent without stimuli (1 point). The total CBA score ranges from 6 to 30, with higher scores indicating better overall cognitive function. Herein, severity was established at 6-point intervals using a criterion of 23 points as mild. The severity levels determined from the CBA total score were categorized as normal (30–29 points), mild (28–23 points), moderate (22–17 points), severe (16–11 points), and most severe (10–6 points).

#### 2.3.3. Functional Independence Measure

The Functional Independence Measure (FIM) [26] is an ADL evaluation table consisting of 18 items that are widely and commonly used in Japan. It comprises motor items related to self-care (6 items), sphincter control (2 items), transfers (3 items), locomotion (2 items), and cognitive items related to communication (2 items) and social cognition (3 items). The level of independence for each item was evaluated on a 7-point scale from 1 (complete assistance) to 7 (complete independence). Motor items were evaluated with 13–91 points, and cognitive items with 5–35 points for a total FIM score of 18–126 points, with higher scores indicating greater ADL independence. We used the total score of 13 FIM motor items (13–91) and the score of each ADL item in the FIM motor items. ADL independence at discharge was assessed using an FIM score of ≥6 (6 = modified independence, 7 = full independence).

### 2.4. Statistical Analysis

Descriptive statistics, including the median and first and third quartiles, were first calculated for all patients’ attributes and assessment results. Then, patients were assigned to five groups according to CBA severity at discharge, and descriptive statistics, including the percentage of independence in each ADL, were calculated. The total scores of the FIM motor items were compared for each group using the Kruskal–Wallis test. The Bonferroni method was used to adjust the significance level for multiple comparisons. Multiple logistic regression analysis was performed to clarify the association between CBA severity and independence in ADL items affecting discharge to home [25] (FIM score on dressing the upper body, transfers to bed/chair/wheelchair, transfers to the toilet, sphincter control-bladder management), walking (FIM score > 6 points). The presence or absence of mild or normal CBA was entered into the regression model. Factors affecting the functional prognosis of patients with stroke [6,27], such as age, sex, post-onset period, and BRS (lower limb stage III or lower or other), were included as covariates. SPSS version 13.0J for Windows (SPSS Japan Inc., Tokyo, Japan) was used for all statistical analyses, and the statistical significance level was set at 5% in all cases.

## 3. Results

### 3.1. Patient Characteristics

Table 1 summarizes the patient characteristics. Patients (age range: 20–97 years) had a median of 118 days since stroke onset and a length of admission of 94 days. The median CBA and FIM motor scores were 24 and 81 points, respectively.

### 3.2. ADL Independence and Severity Levels of Cognitive Function Assessed by the CBA

The severity levels of cognitive function assessed by the CBA were 12 (1.5%), 84 (10.6%), 231 (29.2%), 357 (45.1%), and 107 (13.5%) for the most severe, severe, moderate, mild, and normal groups, respectively (Table 2). None of the independent patients in all ADLs were confirmed to have the most severe disease. “Mild” severity was almost independent in ADLs (eating, 95.5%; dressing the upper body, 91.0%; bladder management, 90.8%; transfers to the bed/chair/wheelchair, 90.5%; transfers to the toilet, 90.2%; and walking, 84.3%) (Table 3). The median FIM motor scores according to CBA severity stage were 14 for the most severe group, 27 for the severe group, 65 for the moderate group, 86 for the mild group, and 90 for the normal group. According to the Kruskal–Wallis test, significant differences were noted in the FIM motor scores. Similarly, significant differences were found between the groups using the Bonferroni method for multiple comparisons (*p* < 0.01) (Figure 2).

### 3.3. Relationships between CBA Severity and Independence in Walking or ADLs Affecting Discharge to Home

Logistic regression analyses revealed that mild or normal levels based on the CBA were associated with higher odds ratios (ORs) for independence in dressing the upper body (OR = 21.90; 95% confidence interval (CI), 13.50–35.70), bladder management (OR = 11.60; 95% CI, 7.21–18.60), transfers to the bed/chair/wheelchair (OR = 18.30; 95% CI, 11.40–29.40), transfers to the toilet (OR = 18.30; 95% CI, 11.40–29.30), and walking (OR = 16.60; 95% CI, 10.60–26.10) in the adjusted model with age, sex, time since onset, and BRS (lower limb stage III, lower, or other) (Table 4).

## 4. Discussion

The total scores of the FIM motor items differed according to the CBA severity stage, and the proportions of independent patients in each ADL were approximately 30% in the moderate CBA group, 90% in the mild CBA group, and almost 100% in the normal CBA group. A CBA severity stage of mild or normal was associated with independence in ADLs that affect discharge to home.

ADL abilities at each CBA severity stage were different. The CBA results were associated with ADL ability and may be suitable to indicate the characteristics of patients with stroke in the recovery stage, including ADLs [20]. Cognitive functions associated with ADLs include consciousness, memory, and attention [28,29,30]. The fact that the CBA also includes these cognitive functions may impact the characterization of ADLs. Other CBA items are designed to interpret the hierarchy of the neuropsychological pyramid and may be suitable for a balanced assessment of cognitive function. Lesniak et al. reported that attention, language, short-term memory, and executive functions were the most frequently affected cognitive impairments in patients with early stroke [31]. Jokinen et al. reported a lower frequency of detectable cognitive impairment on the MMSE than on extensive neuropsychological assessment [32]. These reports suggest that cognitive impairment in patients with stroke should be evaluated comprehensively, considering language disorders. Comprehensive evaluations of cognitive function may favor observational assessments, such as the CBA.

The most severe CBA group required full assistance with ADLs. This group recognized a remarkable lowering in all items, including consciousness and emotion, even in the CBA subitems. Thus, it may be the condition in which the cognitive activity is almost unrecognized, and it could be regarded as a patient group in which it is difficult to draw out the patient’s own activity. Stroke survivors with a more severe disability require more hours of informal care [33]. The hours of informal care required affect the caregiving burden [34]. Impaired consciousness is also associated with complications [35]. Therefore, it is assumed that the home becomes inappropriate as the discharge destination, and medical management is considered necessary to continue life.

The severe CBA group had severely affected ADLs but could still perform some ADLs. The scores of the CBA subitems in this group ranged from 2 to 3 points. The scores of each CBA item were similar to those in a group with low ADL independence in a previous study [20]. A few patients could eat a meal independently in the present study. The severe group is expected to perform ADLs through communication within a limited range because cognitive functions are preserved, unlike the most severe group.

The moderate CBA group showed improved ADLs and the assistance quantity compared to the severe group. Many patients were independent in their eating and could transmit their urination desire; additionally, transfer and walking became partially possible. This group may have mostly improved arousal and spontaneity, and motor learning was enhanced by improved attention and memory. However, there are insufficient points in attention and detailed memory, and there are many cases in which a lack of seriousness is recognized in judgment and disease perception. Thus, it is difficult for them to become independent. Cognitive and emotional adjustment is necessary during recovery to prevent falls in patients with stroke [36]. On the basis of these findings, it can be inferred that these patients need to be compensated through environmental adjustments and involvement to increase awareness of the disease. In addition, ADL ability is recognized from low to high, and life support is required in proportion to each patient’s ADL ability.

The mild CBA group achieved independence in most ADLs. Although the patients had a high degree of attention and memory loss compared with the pre-symptomatic state, they could judge situations to the extent required in daily life; they would be able to make decisions and act on their own in a well-developed environment. Attention must be paid to each person’s individuality, and involvement in social independence is required.

The normal CBA group had a higher degree of independence in ADLs than the mild CBA group; the problematic ADLs in this group were bathing and transferring to the tub. This relates to FIM motor items, where the most difficult ADLs are bathing and transferring to the tub [37]. The percentage of independent persons in the normal group for bathing and transfers to the tub was approximately 80%, suggesting that abilities other than cognitive function are required. A comprehensive neuropsychological assessment in a previous study also revealed that 17% of stroke patients had no cognitive impairment [32]. In the present study, the proportion of patients in the normal CBA group was 14%, with a similar rate of cognitive impairment. Therefore, in the normal CBA group, it is essential to focus on instrumental ADLs. It is recommended that cognitive function is not identified as a problem and that physical function provides a better understanding of the patients as they experience the illness and reintegrate into society.

A CBA severity of more than mild was associated with ADLs and walking required for discharge to home. Cognitive function has also been reported to be relevant in ADLs, including walking, [6,38] and these cognitive impairments are relevant not only in judgments on neuropsychological tests but also in judgments on screening. CBA severity was associated with more than mild severity in the adjusted model. These findings suggest that the CBA can be used as an indicator for judging independence in ADLs. Although the criteria for independence in walking are balance function [39] and walking ability [40], falls after independence are a concern. In fact, cognitive impairment and physical function have been reported as risk factors for falls in patients with stroke [7]. Therefore, we believe it is necessary to consider the state of cognitive function and safe and independent ADLs. We also believe that monitoring the behavior of patients with stroke and assessing their cognitive function using the CBA is useful for understanding their status.

This study has some limitations. There is no doubt that the results of the severity assessment are dependent on the quality and quantity of information collected by the assessor. It is also possible that the characteristics of the judgment differ depending on the differences in time and place observed, such as during the day and night. Therefore, it is advisable to share this information across multiple professions. Given that the CBA does not cover generalized poststroke cognitive impairment, there is also a need to understand individual symptoms based on neuropsychological test results. Another limitation remains in terms of generalization, as our study included patients from a single center, and we analyzed retrospective data. Additionally, the clinical type of stroke was not taken into consideration when classifying the sample. The clinical type of stroke has also been reported as a prognostic factor for ADL in stroke patients [41] and may have influenced the results of the present study. Furthermore, the discharge destination has been determined by doctors, consultants, and other medical staff through interviews with patients and their families; however, it is unclear whether the discharge destination is appropriate. It is, therefore, necessary to conduct a post-discharge survey. Hence, future research should clarify the state of cognitive function at the time of independence and causal relationships through longitudinal studies.

The CBA severity stages exhibited different characteristics in terms of the percentage of patients with independent ADLs and ADL abilities. A CBA severity of more than mild (23 points) was associated with independence in ADLs and walking, which are important for discharge to home. Understanding the criteria of cognitive function required for independence in each ADL and the characteristics of the severity of CBA will provide clues for setting rehabilitation goals and the rehabilitation content required by therapists. Nurses, caregivers, and other professionals involved in patient care may also provide clues. Additionally, sharing images that show the CBA severity of patients among multiple professions will provide high-quality rehabilitation care.

## 5. Conclusions

Our findings suggest that CBA severity stages exhibited different characteristics in terms of the proportion of patients with independent ADLs and ADL abilities. A CBA severity stage of mild or normal may be associated with independence in ADLs, which affects discharge to home. In the future, longitudinal studies are needed to clarify the state of cognitive function at the time of independence in ADLs and examine a combination of predictors, including modifiable factors that may influence the achievement of independence in ADLs.

## Figures and Tables

**Figure 1 ijerph-20-03005-f001:**
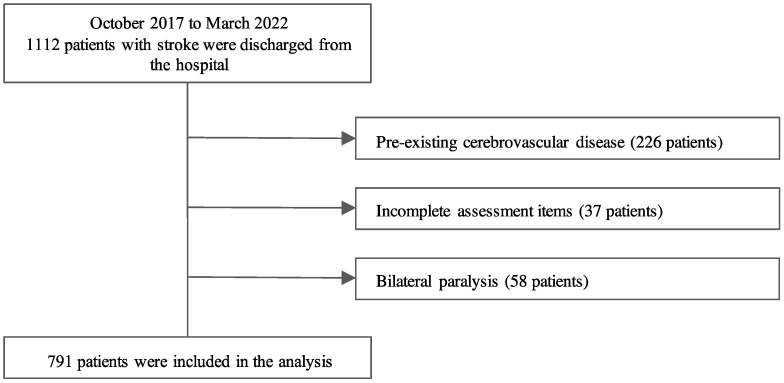
Flow of patients.

**Figure 2 ijerph-20-03005-f002:**
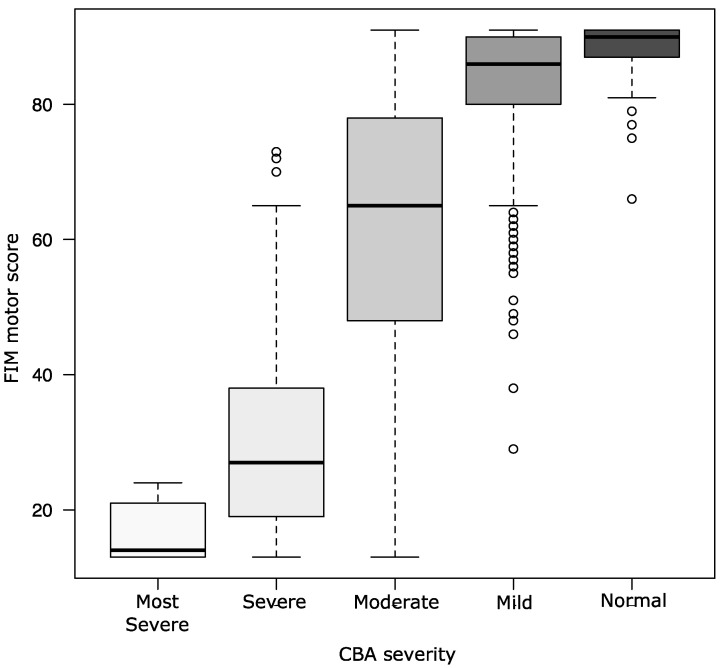
ADL ability score according to CBA severity. Summary data of FIM motor score, according to CBA severity: score of 30–29 indicating normal severity, 28–23 indicating mild severity, 22–17 indicating moderate severity, 16–11 indicating severe severity, and 10–6 indicating most severe severity. Data are represented as boxplots with median, min, and max values. The difference in FIM motor score was observed in all groups (Bonferroni correction to adjust for multiple comparisons). FIM—Functional Independence Measure; CBA—Cognitive-related Behavioral Assessment.

**Table 1 ijerph-20-03005-t001:** Patient characteristics.

	n = 791
Age (years)	73.00 [61.00, 81.00]
Sex, female/male	331/460
Type of stroke, cerebral infarction/cerebral hemorrhage/subarachnoid hemorrhage	448/284/59
Hemisphere of stroke, left/right/other *	378/346/67
Days between stroke onset and admission	24.00 [17.00, 32.00]
Days between stroke onset and discharge	118.00 [80.00, 153.00]
Days of admission	94.00 [57.00, 125.00]
Brunnstrom Recovery Stage, n (%)	
Stage I	8 (1.0)
Stage II	54 (6.8)
Stage III	87 (11.0)
Stage IV	55 (7.0)
Stage V	84 (10.6)
Stage VI	294 (37.2)
None	209 (26.4)
Cognitive-related Behavioral Assessment total score	24.00 [20.00, 27.00]
Cognitive-related Behavioral Assessment severity, n (%)	
Most severe	12 (1.5)
Severe	84 (10.6)
Moderate	231 (29.2)
Mild	357 (45.1)
Normal	107 (13.5)
Functional Independence Measure motor score	81.00 [58.00, 89.00]
Functional Independence Measure cognitive score	30.00 [22.00, 34.00]
Functional Independence Measure total score	111.00 [83.00, 122.00]
Home discharge, n (%)	567 (71.7)

Data are presented as median (interquartile range) or number (percentage). * Other: bilateral strokes and cerebellar/brainstem damage.

**Table 2 ijerph-20-03005-t002:** Patient characteristics by CBA severity.

	Most Severe (n = 12)	Severe (n = 84)	Moderate (n = 231)	Mild (n = 357)	Normal (n = 107)
Age (years)	83.00 [75.25, 88.00]	80.00 [74.00, 85.25]	77.00 [68.00, 83.50]	70.00 [58.00, 78.00]	60.00 [50.00, 74.50]
Sex, female/male	7/5	41/43	89/142	151/206	43/64
Type of stroke, cerebral infarction/cerebral hemorrhage/subarachnoid hemorrhage	7/4/1	52/27/5	131/84/16	194/131/32	64/38/5
Hemisphere of stroke, left/right/other *	9/3/0	39/38/7	111/102/18	170/153/34	49/50/8
Days since stroke onset	150.50 [137.50, 159.25]	150.00 [125.75, 168.00]	134.00 [103.00, 165.00]	107.00 [77.00, 140.00]	74.00 [52.50, 118.00]
Brunnstrom Recovery Stage, n (%)					
Stage I	0 (0.0)	2 (2.4)	2 (0.9)	3 (0.8)	1 (0.9)
Stage II	7 (58.3)	22 (26.2)	21 (9.1)	4 (1.1)	0 (0.0)
Stage III	1 (8.3)	14 (16.7)	37 (16.0)	31 (8.7)	4 (3.7)
Stage IV	2 (16.7)	7 (8.3)	16 (6.9)	24 (6.7)	6 (5.6)
Stage V	2 (16.7)	12 (14.3)	27 (11.7)	34 (9.5)	9 (8.4)
Stage VI	0 (0.0)	15 (17.9)	80 (34.6)	153 (42.9)	46 (43.0)
None	0 (0.0)	12 (14.3)	48 (20.8)	108 (30.3)	41 (38.3)
Cognitive-related Behavioral Assessment total score	9.00 [8.50, 9.00]	14.00 [13.00, 15.00]	20.00 [19.00, 21.00]	26.00 [24.00, 27.00]	30.00 [29.00, 30.00]
Cognitive-related Behavioral Assessment subitem score					
Consciousness	2.00 [2.00, 2.00]	3.00 [3.00, 4.00]	4.00 [4.00, 4.00]	5.00 [4.00, 5.00]	5.00 [5.00, 5.00]
Emotion	2.00 [1.00, 2.00]	3.00 [2.75, 3.00]	4.00 [3.00, 4.00]	5.00 [4.00, 5.00]	5.00 [5.00, 5.00]
Attention	2.00 [1.00, 2.00]	2.00 [2.00, 3.00]	3.00 [3.00, 3.00]	4.00 [4.00, 4.00]	5.00 [4.50, 5.00]
Memory	1.00 [1.00, 1.00]	2.00 [2.00, 2.00]	3.00 [3.00, 4.00]	4.00 [4.00, 5.00]	5.00 [5.00, 5.00]
Judgment	1.00 [1.00, 1.00]	2.00 [2.00, 2.00]	3.00 [3.00, 3.00]	4.00 [4.00, 4.00]	5.00 [5.00, 5.00]
Consciousness of disease	1.00 [1.00, 1.00]	2.00 [1.00, 2.00]	3.00 [3.00, 3.00]	4.00 [4.00, 4.00]	5.00 [5.00, 5.00]
Functional Independence Measure motor score	14.00 [13.00, 20.00]	27.00 [19.00, 38.00]	65.00 [48.00, 78.00]	86.00 [80.00, 90.00]	90.00 [87.00, 91.00]
Functional Independence Measure cognitive score	6.50 [5.00, 8.75]	13.00 [10.00, 15.00]	24.00 [19.50, 27.00]	32.00 [30.00, 34.00]	35.00 [35.00, 35.00]
Functional Independence Measure total score	21.00 [19.00, 29.25]	42.00 [30.00, 51.50]	86.00 [70.50, 103.50]	117.00 [110.00, 123.00]	124.00 [122.00, 126.00]
Home discharge, n (%)	1 (8.3)	16 (19.0)	131 (56.7)	316 (88.5)	103 (96.3)

Data are presented as median (interquartile range) or number (percentage). CBA—Cognitive-related Behavioral Assessment. * Other: bilateral strokes and cerebellar/brainstem damage.

**Table 3 ijerph-20-03005-t003:** Percentage of independent ADLs by CBA severity (Functional Independence Measure motor item).

	Most Severe (n = 12)	Severe (n = 84)	Moderate (n = 231)	Mild (n = 357)	Normal (n = 107)
Self-care					
Eating, n (%)	0 (0.0)	12 (14.3)	163 (70.6)	341 (95.5)	106 (99.1)
Grooming, n (%)	0 (0.0)	2 (2.4)	81 (35.1)	322 (90.2)	106 (99.1)
Bathing, n (%)	0 (0.0)	0 (0.0)	19 (8.2)	189 (52.9)	90 (84.1)
Dress Up, n (%)	0 (0.0)	0 (0.0)	81 (35.1)	325 (91.0)	107 (100.0)
Dress Low, n (%)	0 (0.0)	0 (0.0)	80 (34.6)	325 (91.0)	106 (99.1)
Toileting, n (%)	0 (0.0)	2 (2.4)	87 (37.7)	327 (91.6)	106 (99.1)
Sphincter control					
Bladder, n (%)	0 (0.0)	4 (4.8)	104 (45.0)	324 (90.8)	107 (100.0)
Bowel, n (%)	0 (0.0)	4 (4.8)	97 (42.0)	317 (88.8)	102 (95.3)
Transfers					
Bed Trans, n (%)	0 (0.0)	1 (1.2)	86 (37.2)	323 (90.5)	107 (100.0)
Toilet Trans, n (%)	0 (0.0)	2 (2.4)	82 (35.5)	322 (90.2)	106 (99.1)
Tub Trans, n (%)	0 (0.0)	0 (0.0)	19 (8.2)	170 (47.6)	88 (82.2)
Locomotion					
Walking, n (%)	0 (0.0)	0 (0.0)	62 (26.8)	301 (84.3)	104 (97.2)
Stair climbing, n (%)	0 (0.0)	2 (2.4)	49 (21.2)	236 (66.1)	97 (90.7)

Data are presented as numbers (percentage). Dress Up—dressing the upper body; Dress Low—dressing the lower body; Bed Trans—transfers to the bed/chair/wheelchair; Toilet Trans—transfers to the toilet; Tub Trans—transfers to the tub/shower; and CBA—Cognitive-related Behavioral Assessment.

**Table 4 ijerph-20-03005-t004:** Association of mild or normal level based on the CBA with independence in walking and independence in ADL items affecting discharge to home.

Dependent Variable (FIM Score 6–7)	Independent Variable	FIM Score 6–7, Percentage Applicable	Adjusted Model *	
			OR (95% CI) #	*p*-Value
Dress Up	CBA severity (mild or normal)	432/464 (93.1%)	21.90 (13.50–35.70)	<0.01
Bladder		431/464 (92.9%)	11.60 (7.21–18.60)	<0.01
Bed Trans		430/464 (92.7%)	18.30 (11.40–29.40)	<0.01
Toilet Trans		428/464 (92.2%)	18.30 (11.40–29.30)	<0.01
Walking		405/464 (87.3%)	16.60 (10.60–26.10)	<0.01

# Odds ratios from logistic regression analysis. * Adjusted for age, sex, days since stroke onset, and Brunnstrom Recovery Stage (III or other). OR—odds ratio; FIM—Functional Independence Measure; CI—confidence interval; Dress Up—dressing the upper body; Bed Trans—transfers to the bed/chair/wheelchair; Toilet Trans—transfers to the toilet; and CBA—Cognitive-related Behavioral Assessment.

## Data Availability

The data presented in this study are available on reasonable request from the corresponding author.

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
