# Peer review of "Association between the Cognitive-Related Behavioral Assessment Severity Stage and Activities of Daily Living Required for Discharge to Home in Patients with Stroke: A Cross-Sectional Study"

_ijerph, 2023, doi:10.3390/ijerph20043005_

Round 1

Reviewer 1 Report

This study examined the activities of daily living characteristics according to the severity defined by the CBA total score in 791 stroke survivors. Furthermore, they also investigated the impact of the severity of the CBA on ADL. 

The work is interesting and well-written. The dataset collected and used in this study is large, leading to more robust results.

I have only a few minor comments: 

In section 2 you describe the study design but information about when the assessment is made are missing. For example, the difference between the assessment and even if the assessment is made only once or multiple times for each patient.

In your opinion, the strong difference between the sample size in the different groups is affecting the results? In addition, it was unexpected to see such a large number of people falling in the 'normal' category. You said that this can depend on the evaluation, did you find the literature to support the numerosity of each group?

Author Response

添付ファイルをご覧ください。

Reviewer 2 Report

In this study, the authors make a parallel between patients' cognitive function after stroke, assessed by Cognitive-related Behavioral Assessment and activities of daily living in order to help doctors determine when a patient is able to return home. The paper is well written and easy to read. However, I had a few questions/comments:

Introduction :
- The goal of linking everyday activities with the CBA is interesting but I was wondering if there was a request to develop a new battery of tests from the nursing staff? Are there sometimes patients sent home when they shouldn't have been due to lack of decisional tests?

Materials and methods:
- line 79: "Informed consent was obtained from the patients by an opt-out option..." Form de patient or his legal guardian when needed? Were all patients able to give informed consent? Including those after a severe stroke?
- Line 107: "The CBA is performed by observing the patient's behavior in daily life". Were these observations made at home or in the last days before leaving the hospital? Did the patients in the study all go home already or were they in a rehabilitation ward at the hospital?
- Line 119-121: "The severity levels determined from the CBA total score were categorized as ..." Is this a test developed by your group in reference n°20? The reference could be added here.
- Line 123-124: The functional Independence Measure (FIM)26... here also the test reference could be added. Do you have any idea why the test is called FIM26 and it includes 18 items?

Reviewer 3 Report

Although the article is interesting, it raises many questions

The analysis of groups without taking into account the clinical type of stroke may distort the obtained results,

Based on the available knowledge, comparing both groups together in the assessment of rehabilitation effects, where the initial prognosis in patients after ischemic stroke is worse compared to patients after hemorrhagic stroke, maybe a mistake of choice.

In section 2.1. Ethical considerations add information whether the patient had the right to withdraw from the study?

2.2. Study design and patients add information in the inclusion and exclusion criteria, was age an exclusion criterion, describe the patient's clinical condition according to the NIHSS scale With what score on the scale were the patients included or excluded from the study?

2.4. Statistical analysis

The level of statistical significance was described.

 What power of the statistical test was used?

In the Results section

In both tables 1 and 2 you enter Days since stroke onset 118.00 [80.00, 153.00] – median 3 months

Section 22, How to explain the inclusion criterion of those hospitalized for at least 1 month?

Under the results of the tables, there should be information about the statistical test used

Figure 2. FIM motor score by CBA severity what do the colors of the presented graph mean?

Conclusions should not be a summary of the results carried out, they should be modified

We examined ADL characteristics according to severity, as judged by the CBA total score, in patients with stroke admitted to rehabilitation wards. We also examined the impact of CBA severity on more important ADL items for discharge to home. The results showed that the CBA severity stages exhibited different characteristics in terms of the percentage of patients with independent ADLs and ADL abilities.

Add section limitations. 
